# Exploring Carboxamide Derivatives as Promising Anticancer Agents: Design, In Vitro Evaluation, and Mechanistic Insights

**DOI:** 10.3390/ijms26125903

**Published:** 2025-06-19

**Authors:** Manal M. Al-Najdawi, Maysaa M. Saleh, Dima A. Sabbah, Rima Hajjo, Hiba Zalloum, Suha M. Abudoleh, Duaa A. Abuarqoub, Yusuf M. Al-Hiari, Mohammad Yasin Mohammad, Husam ALSalamat, Hebah Mansour, Nawzat D. Aljbour, Aktham H. Mestareehi

**Affiliations:** 1Department of Pharmaceutical Technology and Cosmetics, Faculty of Pharmacy Middle East University, Amman 11831, Jordan; 2Department of Applied Pharmaceutical Sciences and Clinical Pharmacy, Faculty of Pharmacy, Isra University, Amman 11622, Jordan; 3Department of Pharmacy, Faculty of Pharmacy, Al-Zaytoonah University of Jordan, Amman 11733, Jordan; 4Laboratory for Molecular Modeling, Division of Chemical Biology and Medicinal Chemistry, Eshelman School of Pharmacy, The University of North Carolina, Chapel Hill, NC 27599, USA; 5Advisory Board, Jordan CDC, Amman 11118, Jordan; 6Hamdi Mango Center for Scientific Research, The University of Jordan, Amman 11942, Jordan; 7Department of Basic Pharmaceutical Sciences, Faculty of Pharmacy, Middle East University, Amman 11831, Jordan; 8Department of Pharmaceutical and Biomedical Sciences, Faculty of Pharmacy and Medical Sciences, University of Petra, Amman 11196, Jordan; 9Cell Therapy Center, The University of Jordan, Amman 11942, Jordan; 10Department of Pharmaceutical Sciences, Faculty of Pharmacy, The University of Jordan, Amman 11942, Jordan; 11Pharmacological and Diagnostic Research Center, Faculty of Pharmacy, Al-Ahliyya Amman University, Amman 19328, Jordan; 12Department of Pharmacology, School of Medicine, Al-Balqa Applied University, Al-Salt 19117, Jordan; 13Department of Basic Pharmaceutical Sciences, Faculty of Pharmacy, Isra University, Amman 11622, Jordan; 14Department of Pharmaceutical Sciences, Eugene Applebaum College of Pharmacy and Health Sciences, Wayne State University, Detroit, MI 48202, USA

**Keywords:** carboxamides, cheminformatics, molecular docking, MTT assay, principal component analysis

## Abstract

Carboxamide derivatives are a promising class of compounds in anticancer drug discovery, owing to their ability to interact with multiple oncogenic targets and their favorable pharmacological profiles. In this study, we report the design, synthesis, and biological evaluation of a series of *N*-substituted 1*H*-indole-2-carboxamides as potential anticancer agents. The synthesized compounds were assessed for antiproliferative activity using the MTT assay against MCF-7 (breast cancer), K-562 (leukemia), and HCT-116 (colon cancer) cell lines, with normal human dermal fibroblasts included as a non-cancerous control. Several compounds demonstrated notable cytotoxicity and selectivity. Compounds 12, 14, and 4 exhibited potent activity against K-562 cells, with IC_50_ values of 0.33 µM, 0.61 µM, and 0.61 µM, respectively. Compound 10 showed the most significant activity against HCT-116 cells (IC_50_ = 1.01 µM) with a high selectivity index (SI = 99.4). Moderate cytotoxicity was observed against MCF-7 cells. To elucidate the mechanism of action, molecular docking and induced-fit docking studies were conducted against key cancer-related targets, including topoisomerase–DNA (PDB ID: 5ZRF), PI3Kα (4L23), and EGFR (3W32), revealing favorable binding interactions. Additionally, principal component analysis of molecular descriptors indicated that the compounds possess promising drug-like and lead-like properties, particularly compound 10. Overall, this study highlights N-substituted indole-2-carboxamides as promising scaffolds for further optimization. The integration of synthetic chemistry, biological assays, and computational modeling provides a robust foundation for the continued development of these compounds as potential anticancer agents.

## 1. Introduction

The global burden of cancer is expected to rise dramatically, with cases increasing from 14.1 million in 2012 to 21.6 million in 2023 and projected to reach 28.4 million by 2040 (a 47% surge from 2020). This rapid escalation solidifies cancer as the second leading cause of death worldwide following cardiovascular diseases. In 2020, the most frequently diagnosed cancers included breast, lung, and colorectal cancers (CRC), accounting for approximately 12.5%, 12.2%, and 10.7% of all confirmed cases, respectively [1,2]. Additionally, CRC is the second leading cause of cancer-related deaths globally [2,3]. Leukemia remains the most common cancer in children and continues to contribute significantly to cancer-related morbidity and mortality worldwide [4].

Cancer treatment encompasses a range of therapeutic approaches, including surgery, radiotherapy, hormonal therapies, monoclonal antibody-based treatments, and conventional cytotoxic agents [5,6]. Among these, cytotoxic drugs remain a cornerstone of oncology care, effectively inhibiting cell division and suppressing tumor growth. However, despite significant advancements in anticancer therapy, several challenges persist, including severe adverse effects, limited bioavailability, off-target cytotoxicity, and the development of drug resistance. Overcoming these obstacles necessitates the design and discovery of potent, selective chemotherapeutic agents with enhanced efficacy and reduced toxicity. Leveraging diverse molecular scaffolds and targeting multiple therapeutic pathways are crucial strategies for advancing cancer treatment and achieving these goals [7].

Tyrosine Kinase (TK) Vascular Endothelial Growth Factor Receptor-2 (VEGFR-2) is a cell surface protein that functions as a receptor for growth factors from the vascular endothelial growth factor (VEGF) family [8,9]. These growth factors and their receptors play a pivotal role in regulating blood vessel formation. Among them, VEGFR-2 is the primary mediator of angiogenesis, the process of new blood vessel development [10]. The inhibition of VEGFR-2 has emerged as a promising therapeutic strategy in cancer treatment, driving extensive research efforts in this area [11]. To disrupt the VEGF/VEGFR signaling pathway, various inhibitors have been developed, including small-molecule compounds and monoclonal antibodies [12]. These inhibitors are frequently used in combination with other cancer therapies to enhance treatment efficacy, reduce toxicity, and overcome drug resistance [13]. Notable VEGFR-2 inhibitors include small-molecule drugs such as Sunitinib, Sorafenib, and Pazopanib, as well as monoclonal antibodies like Ramucirumab and Bevacizumab [14,15]. These agents have demonstrated significant clinical promise and have been approved for treating multiple cancer types, including renal cell carcinoma, hepatocellular carcinoma, and colorectal cancer. While VEGFR-2 inhibitors represent a significant advancement in oncology, ongoing research aims to further improve their safety and therapeutic efficacy for better patient outcomes [16].

Topoisomerase inhibitors remain a cornerstone and highly promising in cancer therapy due to their ability to disrupt DNA replication and repair mechanisms, ultimately triggering cancer cell death. Among these, Camptothecin, a well-established topoisomerase I inhibitor, has demonstrated remarkable efficacy in both preclinical and clinical studies [17]. Correspondingly, Etoposide, which targets topoisomerase II, is widely used in the treatment of various malignancies, including small-cell lung cancer, testicular cancer, and ovarian cancer [18,19]. Other clinically approved inhibitors, such as Irinotecan and Topotecan, have also shown significant effectiveness against multiple cancer types [20]. Despite their therapeutic potential, these inhibitors are often associated with adverse effects, including myelosuppression and an increased risk of infections, which can pose challenges to their clinical application.

To address these challenges, ongoing research is focused on developing next-generation topoisomerase inhibitors with enhanced efficacy and reduced toxicity. Efforts are directed toward designing novel structural frameworks and optimizing pharmacokinetic properties to improve therapeutic outcomes. These advancements aim to refine the pharmacological profile of topoisomerase inhibitors while mitigating their associated adverse effects, ultimately improving their clinical applicability [21].

Privileged structures are molecular frameworks known for their versatility, exhibiting diverse biological activities and favorable pharmacokinetic properties. These structures play a crucial role in drug discovery, serving as effective starting points for optimizing structure–activity relationships in new compounds [22]. Their significance has been extensively documented, with examples such as the imidazole scaffold, which has demonstrated antimicrobial, antiviral, and anticancer properties [23]. By leveraging privileged structures, researchers can streamline the drug discovery process, reducing time and resource investment while utilizing well-established molecular frameworks. Furthermore, these structures inspire the design of novel drugs with enhanced activity and specificity. For instance, the benzodiazepine scaffold has contributed to the development of various therapeutics, including anxiolytics, hypnotics, and anticonvulsants, guiding the creation of innovative compounds with improved pharmacological profiles [24].

Furan and indole are well-established privileged structures extensively investigated for their potential in anticancer drug development. Indole, a highly versatile scaffold, plays a crucial role in the design of cancer therapeutics due to its ability to interact with target receptors through hydrogen bonding, hydrophobic interactions, and π–π stacking [25]. Likewise, furan has emerged as a promising framework for anticancer drug discovery, attributed to its diverse pharmacological activities and capacity to form covalent bonds with nucleophilic molecules. The incorporation of an amide bond into indole and furan derivatives has been shown to enhance their binding affinity to target receptors, thereby strengthening their anticancer efficacy [26]. Notably, clinically approved drugs such as Imatinib and Sunitinib, which feature indole or furan scaffolds, have demonstrated significant anticancer activity, further underscoring the therapeutic relevance of these privileged structures [25].

Indole-2-carboxamide derivatives have garnered significant interest in anticancer drug discovery due to their structural versatility and ability to interact with diverse biological targets involved in tumor progression. The indole moiety is a privileged scaffold known for its role in modulating critical signaling pathways such as apoptosis, cell cycle regulation, and topoisomerase inhibition. Moreover, the carboxamide linkage enhances molecular flexibility and offers opportunities for hydrogen bonding, which can improve target affinity and pharmacokinetic properties. In this study, we focused on the synthesis of N-substituted indole-2-carboxamides as potential anticancer agents based on their promising drug-like characteristics and previously reported cytotoxic activities. To further enhance their selectivity and potency, particularly against the MCF-7 cancer cell line, various substituents were introduced to explore structure–activity relationships. Future modifications may include the incorporation of electron-donating or sterically bulky groups, which have been reported to influence cell permeability and target specificity, thereby offering avenues for improved therapeutic performance.

Human cancer cell lines, including HCT-116 (colon cancer), MCF-7 (breast cancer), and K-562 (leukemia), serve as invaluable models in preclinical cancer research, offering critical insights into the biological activity and therapeutic potential of novel anticancer compounds. The MCF-7 cell line, derived from breast cancer and characterized by hormone receptor expression, is widely utilized to study breast cancer biology and evaluate the efficacy of potential therapeutic agents [27].

Research using MCF-7 cells in vitro provides a platform for assessing the mechanisms of action of novel compounds in breast cancer treatment [28]. Likewise, HCT-116, a colorectal cancer cell line, is frequently utilized as a model for studying colorectal cancer and assessing the effectiveness of new therapeutic agents. Data obtained from studies using HCT-116 cells can complement in vitro findings and provide deeper insights into compound activity against colorectal cancer. Additionally, the K-562 cell line, originating from chronic myeloid leukemia (CML), is extensively used to explore CML pathophysiology and assess the potential of emerging treatments. Studies utilizing K-562 cells aid in identifying promising therapeutic candidates and optimizing structure–activity relationships to improve drug efficacy [29]. Collectively, these cancer cell lines play a vital role in evaluating experimental compounds, providing essential insights that contribute to the advancement of bioactive compound revelation and therapeutic improvement.

In computational chemistry research, various proteins are extensively studied to assess the impact of novel compounds on cancers such as breast and colorectal cancers and leukemia. For breast cancer, estrogen receptors (ERs), particularly ERα and ERβ, serve as key targets due to their role in promoting the growth of hormone-dependent breast cancer. In colorectal cancer, proteins such as the epidermal growth factor receptor (EGFR), vascular endothelial growth factor receptor (VEGFR), and components of the mitogen-activated protein kinase (MAPK) signaling pathway are of significant interest, as they play crucial roles in tumor progression and metastasis [30].

Likewise, for leukemia, tyrosine kinase receptors, notably BCR-ABL, are critical therapeutic targets given their involvement in the proliferation and survival of chronic myeloid leukemia (CML) cells [31]. These proteins are essential targets in evaluating the potential of experimental compounds as anticancer agents. Computational techniques such as molecular docking are widely employed to predict the binding affinity and interaction of novel compounds with these target proteins, offering valuable insights into their therapeutic potential [32,33]. These findings contribute to the identification of promising drug candidates and the optimization of their structure–activity relationships, facilitating the advancement of targeted cancer therapies.

The integration of in vitro assays and molecular modeling techniques, such as docking studies, provides critical insights into the anticancer potential of indole carboxamide derivatives. The MTT assay, a widely used method for evaluating antiproliferative activity, enables the assessment of these compounds against various cancer cell lines. Testing synthesized derivatives on HCT-116, MCF-7, and K-562 cell lines yield essential data on their efficacy and potential mechanisms of action. These findings play a pivotal role in guiding further investigations aimed at optimizing the structure–activity relationship (SAR) of indole carboxamide derivatives, thereby advancing their development as promising candidates for anticancer therapeutics.

This study aims to investigate the anticancer potential of indole carboxamide derivatives against MCF-7, colon HCT-116, and K-562 cancer cell lines, with a particular emphasis on understanding their mechanisms of action. Through a comprehensive evaluation of their cytotoxicity, selectivity, and molecular interactions, the study seeks to establish a detailed structure–activity relationship (SAR) that will aid in the rational design of more potent and selective anticancer agents. Additionally, advanced computational modeling, including molecular docking and pharmacokinetic profiling, will be employed to predict binding affinities and optimize lead compounds for further preclinical development. The ultimate goal is to contribute to the development of novel therapeutic agents with enhanced efficacy and reduced toxicity, addressing the challenges of drug resistance and adverse effects associated with current treatments.

## 2. Results and Discussion

The synthetic approach described in this study demonstrates the efficient preparation of a series of *N*-substituted 1*H*-indole-2-carboxamides, featuring diverse aromatic moieties appended to the indole core. This synthetic strategy involved the initial synthesis of the key intermediate, indole-2-carbonyl chloride (2), which acts as a versatile precursor for the subsequent derivatization steps, facilitating the incorporation of various substituents. Indole-2-carbonyl chloride (2) was synthesized by refluxing indole-2-carboxylic acid with thionyl chloride (SOCl_2_) in the presence of a catalytic amount of *N*,*N*-dimethylformamide (DMF). This reaction efficiently converted the carboxylic acid group into the corresponding acid chloride, yielding a notably high conversion rate of 84%. The structural identity of the product was confirmed through characteristic spectroscopic data, ensuring the successful formation of the desired intermediate.

The acid chloride (2) served as a key building block for the synthesis of a diverse serious of *N*-substituted 1*H*-indole-2-carboxamides. The derivatization reactions proceeded efficiently, yielding the target compounds with reasonable to excellent yields (32% to 71%). These transformations were achieved through the reaction of acid chloride (2) with various aromatic amines under optimized conditions, as seen in Figure 1, leading to the successful formation of the desired carboxamide derivatives.

Compound 16, *N*-(9,10-Dihydro-9,10-dioxoanthracen-1-yl)-2-furamide, was synthesized through the reaction of 1-aminoanthraquinone with furan-2-carbonyl chloride. This transformation proceeded smoothly and efficiently and yielded carboxamide 16 as a dark red solid, indicating its successful synthesis as shown in Figure 2. The structural elucidation was supported by comprehensive spectroscopic analysis.

In conclusion of these syntheses, this study establishes an efficient and versatile synthetic strategy for the preparation of *N*-substituted 1*H*-indole-2-carboxamides with diverse aromatic functionalities, including the structurally distinctive compound 16. The successful synthesis and characterization of these derivatives highlight their potential as promising candidates for further investigation in drug discovery, particularly in the development of novel anticancer agents and other biologically relevant molecules.

The proliferation inhibitory activities of the carboxamide derivatives 4-16 were tested in vitro using the MTT assay [34,35] against the human cancer MCF-7, K-562, and HCT-116 cell lines in addition to normal human cells (dermal fibroblasts) to evaluate their putative selective cytotoxic activities. The concentrations at 50% cell proliferation inhibition (IC_50_) were determined from the dose–response curves, as illustrated in Figure 1, after 72 h exposure of the cells to carboxamide derivatives 4-16 and are presented in Table 1.

We have elaborated on the rationale behind selecting electron-withdrawing groups (EWGs) such as Cl, F, and NO_2_ for compounds 3, 5, and 7. These substituents were chosen based on their known ability to enhance binding interactions with biological targets through increased polarity and potential hydrogen bonding. Specifically, EWGs can influence the electronic distribution within the molecule, potentially improving interactions with active sites of oncogenic proteins. However, we acknowledge the importance of exploring the effects of electron-donating groups (EDGs) like CH_3_ or OCH_3_ on anticancer activity. EDGs can alter the electron density of the molecule differently, which may affect binding affinity and selectivity toward cancer cells. We have highlighted this as a limitation of the current study and propose it as a direction for future research. Furthermore, we have analyzed the differences in anticancer activity among the synthesized compounds, correlating the nature of the substituents with their observed cytotoxic effects. This analysis provides insights into how electronic properties of substituents influence biological activity, thereby guiding the design of more potent anticancer agents in subsequent studies.

The majority of the carboxamide compounds exhibited significant antiproliferative activity against the three carcinoma cell lines. Notably, the K-562 cell line demonstrated the highest sensitivity to carboxamides 12 (featuring a 1-anthraquinone moiety), 14 (with a 2-anthraquinone moiety), and 4 (bearing a *p*-chlorobenzene group), with sub-micromolar IC_50_ values of 0.33 µM, 0.61 µM, and 0.61 µM, respectively. The HCT-116 and MCF-7 cell lines displayed moderate sensitivity, with IC_50_ values ranging from 2.64 to 3.98 µM and 7.16 to 10.6 µM, respectively. Interestingly, the K-562 cell line exhibited considerable resistance to the antiproliferative effects of carboxamides 8 (featuring a *p*-nitrobenzene moiety), 10 (bearing a pyridine ring), and 16 (containing a furyl group), with IC_50_ values exceeding 100 µM, indicating weaker anticancer potential, and may require structural modifications for improved efficacy. Additionally, the MCF-7 cell line demonstrated exceptional resistance to carboxamide 8.

The pyridinyl carboxamide 10 exhibited the most potent antiproliferative activity against the HCT-116 cell line, with an IC_50_ = 1.01 µM and a high selectivity index (SI = 99.4), suggesting strong specificity toward colon cancer cells. Meanwhile, the anthraquinone-based carboxamides 12 (IC_50_ = 0.33 µM) and 14 (IC_50_ = 7.16 µM) demonstrated the highest proliferation inhibitory effects against the MCF-7and K-562 cell lines, respectively. Notably, compounds 10 and 16 displayed selective and potent antitumor activity against HCT-116, showing 69.6- and 10.6-fold selectivity over MCF-7, respectively. In contrast, the *p*-fluorobenzene carboxamide 6 exhibited the lowest antitumor activity against both the MCF-7 and K-562 cell lines, with IC_50_ values of 80.8 µM and 84.6 µM, respectively. Similarly, the weakest antiproliferative effect against the HCT-116 cell line was observed for *p*-nitrobenzene carboxamide 8, with an IC_50_ value of 32.0 µM.

The results revealed that the seven synthesized carboxamides (4-16) exhibited notable selectivity towards cancer cells compared to normal fibroblasts (IC_50_ > 100 µM), with selectivity indices (SIs) ranging from 3.1 to 99.4 (fibroblasts/colon cancer), 1.0 to 14.0 (fibroblasts/breast cancer), and 1.0 to 303 (fibroblasts/leukemia). In particular, carboxamides 12, 14, and 4 demonstrated the highest antitumor activities against K-562 cells, showing the best safety margins with selectivity indices (SIs) of 303 and 164, respectively. Additionally, the MTT assay results revealed that carboxamides 4-16 exhibit significant and selective inhibition of cell proliferation, as well as potent cytotoxic effects against a variety of tumor cell lines. These findings highlight their potential as promising therapeutic candidates for the treatment of several types of carcinomas, including leukemia and colon and breast cancers, suggesting they could be developed into effective anticancer agents.

To explain the anticancer activity of the verified compounds 4-16, we utilized the human topoisomerase–DNA (PDB ID: 5ZRF) complex (resolution = 2.30 Å) [36], phosphoinositide 3-kinase (PI3Kα) (PDB ID: 4L23) (RES = 2.50 Å) [37], and epidermal growth factor receptor (EGFR) (PDB ID: 3W32) (RES = 1.80 Å) complex [38], all obtained from the Protein Data Bank (PDB). These structures were analyzed to understand the molecular interactions between ligands 4–16 and the binding sites of 5ZRF, 4L23, and 3W32. Additionally, to explore the complexes of 5ZRF, 4L23, and 3W32/ligands 4-16 in the binding sites, we carried out induced-fit docking (IFD) [39,40,41,42] studies against 5ZRF, 4L23, and 3W32. The IFD results revealed that ligands 4-16 occupy the binding domains enclosed by topoisomerase residues and DNA nucleotide backbones (Figure 2A, Figure 3A and Figure 4A), as well as the PI3Kα and EGFR binding sites. Notably, the scaffold of ligand 16 closely resembles the template of co-crystalized ligands EVP (in 5ZRF), X6K (in 4L23), and W32 (in 3W32) (Figure 2B, Figure 3B and Figure 4B).

The IFD explores the conformational changes in the coordinates of 5ZRF, 4L23, and 3W32. Ligands are docked into the binding domains of these structures using Glide docking [39,40,41,42], followed by energetic minimization of the top ligand geometry along with the binding sites of 5ZRF, 4L23, and 3W32 using the Prime algorithm [39]. A redocking protocol is then applied to the minimized conformations of 5ZRF, 4L23, and 3W32. The flexibility of these proteins is assessed throughout the docking process. The synthesized carboxamide derivatives bind to the 5ZRF backbone, particularly to residues R503, Q778, DG7, DC8, DA12, and DG13, forming hydrogen bonds, as shown in Table 2. Furthermore, recorded computational [43,44,45,46] and experimental [36] data highlight the contribution of Q778, DG13, and DC8 in the interaction with the topoisomerase II-DNA complex. The docking data revealed that the synthesized derivatives bind with PI3Kα residues, specifically W780, E798, Y836, E849, V851, S854, and Q859, as illustrated in Table 3. Reported computational [47,48,49,50,51,52,53,54,55,56,57,58,59,60,61] and experimental [37,62] studies highlight the significance of W780, E798, Y836, E849, V851, S854, and Q859 in PI3Kα complex formation. Moreover, the derivatives bind with EGFR residues, distinctively K745, M793, T854, and D855 (Table 4). Disclosed computational [51,63,64,65] and experimental [38] studies emphasize the significance role of K745, M793, T854, and D855 in the formation of the EGFR complex.

It is important to note that a more negative binding score indicates a better binding affinity. The docking results against 5ZRF show that the three fused rings in ligands 12, 14, and 16 enhance the binding affinity. This finding aligns with previous studies [45,46] and recommends that incorporating at least four fused rings (like in Doxorubicin) or three fused rings combined with bulky or branched alkyl groups (such as in Mitoxantrone) is key to eliciting activity. Interestingly, tailoring the scaffold with *p*-NO_2_ (ligand 8) demonstrates different binding modes, suggesting that the rigidity of NO_2_ might help orient the ligand more effectively in the binding cleft. Overall, the results suggest that a system with three or four fused rings, coupled with a rigid moiety, is necessary for inducing activity. Specifically, DNA intercalation requires planar, rigid, and fused rings to disrupt the H-bond between DNA strands or to intercalate between them. Docking results for PI3Kα reveal that ligands 12 and 16 exhibit comparable affinities to the X6K co-crystallized ligand. However, the docking studies against EGFR indicate a better binding affinity, anticipating that derivatives may be selective for EGFR. In fact, ligands 8, 12, 14, and 16 show comparable binding affinities to W32.

To evaluate the performance of the IFD program, we compared the docked conformations of EVP in 5ZRF, X6K in 4L23, and W32 in 3W32 with their native poses in the crystal structures, as seen in Figure 5. The RMSD (Root Mean Square Deviation) for heavy atoms (all atoms except hydrogen) of EVP, X6K, and W32 between the IFD-generated poses and their native orientations were 0.1403, 0.1131, and 0.4937 Å, respectively. These results indicate that the IFD program is capable of accurately predicting the native pose within the crystal structure and effectively defining the ligand binding conformation.

All 2D AlvaDesc molecular descriptors [66] were calculated for the synthesized molecules 4-16 (Appendix A). These descriptor values were subsequently analyzed using principal component analysis (PCA). The PCA results showed that the molecules clustered into four main groups within the 2D space defined by the first two principal components, which explained the variance in the descriptors, as illustrated in Figure 6A. This clustering demonstrated that the synthesized compounds covered a broad range of physicochemical properties while maintaining acceptable drug-like and lead-like scores. Further evaluation using eight drug-like and two lead-like molecular descriptors from Kode Cheminformatics [66] indicated that compound 10 emerged as the most promising molecule, exhibiting the best drug-like and lead-like properties, as illustrated in the radar plot (Figure 6B). All drug-like scores (DLSs) and lead-like scores (LLSs) ranged from 0 to 1, where higher scores indicate greater similarity to the properties of actual drugs and leads.

## 3. Materials and Methods

### 3.1. Synthesis of Target Compounds and Materials

All chemicals, reagents and solvents were purchased from commercial sources ((SigmaAldrich (St. Louis, MO, USA), Riedel-de Haen (Seelze, Lower Saxony, Germany), Fluka (Buchs, St. Gallen, Switzerland), Ubichem (Budapest, Hungary), Aldrich (Milwaukee, Wisconsin, USA), Tedia (Fairfield, OH, USA), and Merck Corporation (Darmstadt, Hesse, Germany)) and were used in the experimentation without further purification. 1*H*-indole-2-carboxylic acid, 1*H*-furoyl-2-carboxylic acid, 2-aminopyridine, *p*-chloroaniline, *p*-fluoraniline, 1-aminoanthraquinone, phenyl magnesium chloride, thionyl chloride, and silica gel were bought from (Aldrich, USA). *p*-Nitroaniline, hydrochloric acid and potassium carbonate were bought from (Riedel-De Haen, Germany). Benzene, chloroform, ethanol, acetone, dimethyl sulfoxide, ethyl acetate, cyclohexane, pyridine, methanol, and dichloromethane were bought from (Tedia, USA). Dimethylformamide was bought from (Merck, Germany). Magnesium sulfate anhydrous was bought from (Scharlau, Spain). Filter paper was bought from (Whatman, England).

### 3.2. Spectroscopic and Characterization Techniques

Nuclear Magnetic Resonance (NMR) spectra were recorded using a Varian Oxford-300 (300 MHz) spectrometer (Palo Alto, CA, USA) at the Faculty of Pharmacy, University of Jordan. Chemical shifts are reported in parts per million (ppm) relative to tetramethylsilane (TMS) as an internal standard. Deuterated dimethyl sulfoxide (DMSO-*d_6_*) and deuterated chloroform (CDCl_3_) were used as solvents for sample preparation, unless otherwise specified. ^1^H NMR data are presented in the following format: chemical shift (ppm), multiplicity, coupling constant (*J* value) in hertz (Hz), number of protons, and the corresponding proton (s).

Infrared (IR) spectra were recorded using a Shimadzu 8400F FT-IR spectrophotometer (Kyoto, Japan) at the Faculty of Pharmacy, Al-Zaytoonah Private University. Samples were dissolved in chloroform (CHCl₃) and analyzed as thin solid films using a potassium bromide (KBr) disc (Merck, Darmstadt, Germany).

Melting points (*m.p.*) were determined using a Stuart Scientific Electrothermal Melting Point Apparatus (CA, USA) at the Faculty of Pharmacy, University of Jordan. Measurements were conducted in open capillaries and are reported as uncorrected values.

High-resolution mass spectra (HRMS) were acquired in positive or negative ion mode using the ElectroSpray Ionization (ESI) technique with collision-induced dissociation on a Bruker APEX-4 (7 Tesla), (Billerica, MA, USA) instrument at the Department of Chemistry, University of Jordan. Samples were dissolved in acetonitrile, diluted in spray solution (methanol/water 1:1 *v*/*v* + 0.1% formic acid), and infused via a syringe pump at a flow rate of 572 µL/min. External calibration was performed using an arginine cluster within the mass range *m*/*z* 175–871.

Elemental analysis (EA) of carbon (C), hydrogen (H), and nitrogen (N) was conducted using a Euro Elemental Analyzer (Milan, Italy) at the Faculty of Pharmacy, University of Jordan. Moreover, thin-layer chromatography (TLC) was performed on aluminum plates pre-coated with fluorescent silica gel GF254 (ALBET, Bremen, Germany). Silica gel (70–230 Mesh) from GCC (London, United Kingdom) was used for preparative TLC. The plates were visualized under a UV lamp using a Spectroline cabinet, Model CX-20 (Westbury, NY, USA).

### 3.3. Synthesis of Target Compounds and Data Analysis

Compound (**2**), Indole-2-carbonyl chloride

In a round-bottomed flask, 25 mL of anhydrous chloroform (CHCl_3_) was added, followed by the addition of indole-2-carboxylic acid (1 g, 6.2 mmol), thionyl chloride (3.6 mL, 50 mmol), and a small quantity of *N*,*N*-dimethylformamide (DMF). The reaction mixture was subjected to reflux conditions, maintaining a temperature within the range of 70 to 80 °C for a duration of 18 h to ensure the reaction was complete. Following the reaction, the excess thionyl chloride (SOCl_2_) was removed by suction, and the solvent was evaporated, resulting in the formation of acid chloride (**2**, 0.95 g, 84%) as a greenish-yellow crystalline precipitate. The product was characterized with an R_*f*_ value of 0.86 (cyclohexane/ethyl acetate, 7:3) and melting point of 115–120 °C. The ^1^H-NMR (300 MHz, DMSO-*d_6_*) showed the following peaks: δ = 11.90 (s, 1H, NH-indole), 7.65 (d, *J* = 7.8 Hz, 1H, H-7), 7.47 (d, *J* = 8.1 Hz, 1H, H-4), 7.26 (dd, *J* = 7.2, 7.8 Hz, 1H, H-6), 7.16 (s, 1H, H-3), 7.07 (dd, *J* = 7.5, 7.2 Hz, 1H, H-5) ppm (Hopkins et al., 2006) [67].

Compound (**4**), *N*-(4-Chlorophenyl)-1*H*-indole-2-carboxamide

A controlled addition of *p*-chloroaniline (0.41 g, 3.22 mmol) in 15 mL of dichloromethane (DCM) was carefully carried out into a solution containing indole-2-carbonyl chloride (**2**, 0.40 g, 2.23 mmol) and triethylamine (TEA, 1.52 g, 15.00 mmol). The reaction mixture was gently stirred at room temperature for 3 h. Following this, the solution was diluted with 15 mL of 1 N hydrochloric acid (HCl), leading to the formation of distinct liquid phases. The organic layer was successively washed with 15 mL of 2 N sodium hydroxide (NaOH) solution and 15 mL of brine (saturated sodium chloride solution). The solvents were then removed under reduced pressure, and the resulting residue was recrystallized from a mixture of methanol (CH_3_OH) and chloroform (CHCl_3_), yielding an off-white crystalline product (**4**, 0.36 g, 60%). The compound was extensively characterized, with the following analytical data providing comprehensive details on its structure and properties:▪R_*f*_ value: 0.60 (cyclohexane/ethyl acetate, 6:4).▪Melting point: 259 °C.▪^1^H -NMR (300 MHz, DMSO-*d_6_*) showed the following peaks, δ ppm: 11.79 (br s, 1H, NH-indole), 10.35 (s, 1H, NHCO), 7.86 (d, *J* = 9.0 Hz, 2H, Ar-H), 7.69 (d, *J* = 8.1 Hz, 1H, Ar-H), 7.49 (s, 1H, Ar-H), 7.47–7.39 (m, 3H, Ar-H), 7.23 (dd, *J* = 8.1, 7.2 Hz, 1H, Ar-H), 7.08 (dd, *J* = 7.2, 7.8 Hz, 1H, Ar-H).▪^13^C -NMR (DMSO-*d_6_*, δ ppm): 160.26 (CONH), 138.43, 137.36, 131.66, 129.09 (C-3′, 5′), 127.57, 127.44, 124.37 (CH-Ar), 122.26 (CH-Ar), 122.05 (C-2′, 6′), 120.44 (CH-Ar), 112.88 (CH-Ar), 104.60 (CH-Ar).▪IR (thin film, cm^−1^): 3406 (NH-indole), 3318 (NHCO), 1651 (CO), 1593, 1539, 1493, 1400, 1315, 1238, 1192, 1099, 1015, 937, 826, 748.▪Elemental analysis (C_15_H_11_N_2_OCl):▪Theoretical values (calculated): C, 66.55%; H, 4.10%; N, 10.35%.▪Experimental values (found): C, 66.40%; H, 3.92%; N, 10.12%.

Compound (**6**), *N*-(4-Florophenyl)-1*H*-indole-2-carboxamide

A blend of *p*-fluoroaniline (**5**, 0.35 g, 2.12 mmol) and triethylamine (TEA, 1.52 g, 15.00 mmol) was meticulously added dropwise to a solution containing indole-2-carbonyl chloride (**2**, 0.38 g, 2.12 mmol) in dichloromethane (DCM, 15 mL). The resulting mixture was stirred at room temperature (RT) for 3 h, after which it was diluted with 15 mL of 1 N hydrochloric acid (HCl), leading to the formation of distinct liquid layers. The organic layer was then sequentially washed with 15 mL of 2 N sodium hydroxide (NaOH) solution and 15 mL of brine. Any residual moisture was removed using anhydrous magnesium sulfate. The solvent was evaporated under reduced pressure, and the resulting crude product was recrystallized from a mixture of methanol (CH₃OH) and chloroform (CHCl₃), affording the *title compound* (**6**, 0.25 g, 46%) as a faint yellow solid. The compound was characterized by the following analytical data:▪R_*f*_ value: 0.54 (cyclohexane/ethyl acetate, 7:3).▪Melting point: 245 °C (decomposed).▪^1^H -NMR (300 MHz, DMSO-*d_6_*, δ ppm): 11.78 (br s, 1H, NH-indole), 10.80 (s, 1H, NHCO), 7.89–7.79 (m, 2H, Ar-H), 7.67 (d, *J* = 7.8 Hz, 1H, Ar-H), 7.48 (d, *J* = 8.0 Hz, 1H, Ar-H), 7.44–7.39 (m, 1H, Ar-H), 7.28–7.15 (m, 3H, Ar-H), 7.11–7.01 (m, 1H, Ar-H).▪^13^C-NMR (DMSO-*d_6_*, δ ppm, rotamers observed): 160.28, 157.09 (d, 1*J* C-F = 283 Hz, C-4′, CH-Ar), 160.14 (CONH), 157.09, 137.30, 135.82, 131.82, 127.47, 124.25 (CH-Ar), 122.44, 122.19 (d, 2*J* C-F = 19 Hz, C-3′, 5′, CH-Ar), 122.34 (CH-Ar), 120.38 (CH-Ar), 115.89, 115.59 (d, 3*J* C-F = 22.13 Hz, C-2′, 6’, CH-Ar), 112.86 (CH-Ar), 104.40 (CH-Ar).▪IR (thin film, cm^−1^): 3422 (NH-indole), 3337 (NHCO), 3271, 1639 (CO), 1543, 1508, 1408, 1339, 1315, 1223, 1188, 1096, 833, 818, 748.▪HRMS (ESI, negative mode): *m*/*z* (M^+^-H^+^) 253.07826 (C_15_H_10_N_2_OF) requires 253.07772.▪Elemental analysis (C_15_H_11_N_2_OF):○Theoretical values: C, 70.86%; H, 4.36%; N, 11.02%.○Experimental values: C, 70.97%; H, 4.15%; N, 10.90%.

Compound (**8**), *N*-(4-Nitrophenyl)-1*H*-indole-2-carboxamide

A solution of indole-2-carbonyl chloride (**2**, 0.45 g, 2.51 mmol) in dichloromethane (DCM, 15 mL) was gradually added to a mixture containing *p*-nitroaniline (**7**, 0.45 g, 3.26 mmol) and triethylamine (TEA, 1.52 g, 15.00 mmol). The mixture was stirred at room temperature for 3 h and then diluted with 15 mL of 1 N hydrochloric acid (HCl), resulting in the formation of two distinct liquid phases. The organic layer was sequentially washed with 15 mL of 2 N sodium hydroxide (NaOH) solution and 15 mL of brine, followed by drying with anhydrous magnesium sulfate. The solvent was removed under reduced pressure, and the crude product was recrystallized from a mixture of methanol (CH₃OH) and chloroform (CHCl₃), yielding the *title compound* (**8**, 0.50 g, 71%) as a green-brown solid. The characterization data of the compound are listed below:▪R_*f*_ value: 0.47 (CHCl_3_/CH_3_OH/CH_3_CO_2_H, 94:5:1).▪Melting point: > 290 °C (decomposed).▪^1^H-NMR (300 MHz, DMSO-*d_6_*, δ ppm): 11.84 (br s, 1H, NH-indole), 10.74 (br s, 1H, NHCO), 8.28 (d, *J* = 9.2 Hz, 2H, H-3′, 5′, Ar-H), 8.11 (d, *J* = 9.2 Hz, 2H, H-2′, 6′, Ar-H), 7.71 (d, *J* = 8.0 Hz, 1H, Ar-H), 7.58–7.45 (m, 2H, Ar-H), 7.76 (dd, *J* = 7.4, 7.5 Hz, 1H, Ar-H), 7.09 (dd, *J* = 7.6, 7.3 Hz, 1H, Ar-H).▪^13^C-NMR (DMSO-*d_6_*, δ ppm): 160.67 (CONH), 145.84, 142.79, 137.68, 131.15, 127.38, 125.31 (CH-Ar), 124.80 (CH-Ar), 122.45 (CH-Ar), 120.92 (CH-Ar), 120.02 (CH-Ar), 112.97 (CH-Ar), 105.69 (CH-Ar).▪Elemental analysis (C_15_H_11_N_3_O_3_):○Theoretical values: C, 64.05%; H, 3.94%; N, 14.94%.○Experimental values: C, 63.97%; H, 3.89%; N, 14.78%.

Compound (**10**), *N*-(Pyridin-2-yl)-1*H*-indole-2-carboxamide

Indole-2-carbonyl chloride (**2**, 0.42 g, 2.34 mmol) was combined with 2-aminopyridine (**9**, 0.42 g, 4.51 mmol) in a flask containing 25 mL of *N*,*N*-dimethylformamide (DMF) and 2.0 mL of triethylamine (TEA, 7.5 mmol). The reaction mixture was heated at 150 °C for 2 days. Afterward, the solvent was removed under reduced pressure, and the residue was dissolved in chloroform (CHCl_3_) and extracted with acidic water. The organic layer was dried over anhydrous magnesium sulfate (MgSO_4_), and the solvent was evaporated. The product was further purified by column chromatography, using chloroform as the eluent, yielding the *target compound* (**10**, 180 mg, 32%) as a faint yellow solid. The compound was thoroughly characterized, and the results of the analyses are summarized below:▪R*_f_*: 0.53 (CHCl_3_/CH_3_OH/CH_3_CO_2_H, 94:5:1).▪Melting point: 260 °C (decomposed).▪^1^H-NMR (300 MHz, DMSO-*d_6_*): rotamers, δ = 13.75 (br s, 1H, NH-indole), 11.8 + 10.91 (2 br s, 1H, NHCO), 8.61–7.78 (m, 3H, Ar-H), 7.75–7.35 (m, 3H, Ar-H), 7.30–6.80 (m, 3H, Ar-H) ppm.▪^13^C-NMR (DMSO-*d_6_*): δ = 160.11 (CONH), 144.10, 137.60, 137.24, 133.32, 127.56, 124.76, 113.56, 112.45, 111.90, 108.37, 104.99.▪IR (thin film): ν = 3449 (NH-indole), 3422 (NHCO), 3144, 2924, 1701 (CO), 1558, 1450, 1366, 1346, 1200, 1080, 748 cm^−1^.▪HRMS (ESI, negative mode): *m*/*z* (M^+^-H^+^) 236.08294 (C14H11N3O) requires 236.08239.▪Elemental analysis (C_14_H_11_N_3_O):○Theoretical values (calculated): C, 70.87%; H, 4.67%; and N, 17.71%.○Experimental values (found): C, 70.62%; H, 4.88%; and N, 17.55%.

Compound (**12**), *N*-(9,10-Dihydro-9,10-dioxoanthracen-1-yl)-1*H*-indole-2-carboxamide

A mixture of 1-aminoanthraquinone (**11**, 0.40 g, 1.8 mmol) and indole-2-carbonyl chloride (**2**, 0.40 g, 2.2 mmol) was refluxed at 120 °C for 18 h using an air condenser. After completion, 15 mL of 1,4-dioxane was added, and the reaction mixture was stirred at room temperature for an additional 24 h. This procedure was adapted from Al-Najdawi et al. [68]. The product was isolated by filtration, washed with ethanol, dried, and purified through recrystallization from chloroform, yielding the *title compound* (**12**, 0.30 g, 45%) as a brown-green solid. The compound was characterized, and the following analytical data were obtained:▪R*_f_*: 0.59 (cyclohexane/ethyl acetate, 6:4).▪Melting point: > 350 °C (decomposed).▪^1^H-NMR (300 MHz, DMSO-*d_6_*) rotamers: δ = 13.23 (br s, 1H, NH-indole), 12.08 (s, 1H, NHCO), 9.15 + 9.13 (2s, 1H, Ar-H), 8.50–7.76 (m, 5H, Ar-H), 7.62–6.89 (m, 5H, Ar-H) ppm.▪^13^C-NMR (DMSO-*d_6_*): δ = 182.60 (CO-ketone), 176.31 (CO-ketone), 163.21 (CONH), 134.37 (CH-Ar), 133.90 (CH-Ar), 132.90, 131.68, 129.37, 128.06 (CH-Ar), 127.69, 127.13, 126.48 (CH-Ar), 125.25, 124.47, 123.85 (CH-Ar), 122.37 (CH-Ar), 121.83, 120.90 (CH-Ar), 118.16 (CH-Ar), 116.20 (CH-Ar), 115.24 (CH-Ar), 112.94 (CH-Ar), 104.18 (CH-Ar) ppm.▪IR (thin film): ν = 3314 (NH-indole), 3213 (NHCO), 3125, 1667 (br CO), 1636 (CONH), 1578, 1531, 1412, 1312, 1269, 1238, 1173, 1015, 806, 741, 705 cm⁻¹.▪HRMS (ESI, negative mode): *m*/*z* (M^+^-H^+^) 365.09317 (C_23_H_13_N_2_O_3_) requires 365.09262.▪Elemental analysis (C_23_H₁_4_N_2_O_3_):○Theoretical values (calculated): C, 75.40%; H, 3.85%; and N, 7.65%.○Experimental values (found): C, 75.19%; H, 4.01%; and N, 7.96%.

Compound (**14**), *N*-(9,10-Dihydro-9,10-dioxoanthracen-2-yl)-1*H*-indole-2-carboxamide

A mixture of 2-aminoanthraquinone (**13**, 0.45 g, 2.0 mmol) and indole-2-carbonyl chloride (**2**, 0.50 g, 2.8 mmol) was refluxed at 120 °C for 18 h using an air condenser. After completion, 15 mL of 1,4-dioxane was added, and the reaction mixture was stirred at room temperature for an additional 24 h. The resulting product was isolated by filtration, thoroughly washed with ethanol, dried, and further purified via recrystallization from chloroform, yielding the *title compound* (**14**, 0.46 g, 63%) as a pale green solid. The compound was thoroughly characterized, and the following analytical data were recorded:▪R*_f_*: 0.41 (cyclohexane/ethyl acetate, 6:4).▪Melting point: > 350 °C (decomposed).▪^1^H-NMR (300 MHz, DMSO-*d_6_*): δ = 11.88 (br s, 1H, NH-indole), 10.79 (s, 1H, NHCO), 8.68 (d, *J* = 3.0 Hz, 1H, Ar-H), 8.39 (d, *J* = 9.0 Hz, 1H, Ar-H), 8.31–8.11 (m, 3H, Ar-H), 8.0–7.83 (m, 2H, Ar-H), 7.75 (d, *J* = 6.0 Hz, 1H, Ar-H), 7.56 (s, 1H, Ar-H), 7.46 (d, *J* = 6.0 Hz, 1H, Ar-H), 7.37–7.03 (m, 2H, Ar-H) ppm.▪^13^C-NMR (DMSO-*d_6_*): δ = 183.0 (CO-ketone), 181.86 (CO-ketone), 160.67 (CONH), 145.19, 137.65, 135.03 (CH-Ar), 134.68 (CH-Ar), 134.58, 133.67, 131.20, 128.89 (CH-Ar), 128.50, 127.43, 127.21 (CH-Ar), 127.13 (CH-Ar), 124.94 (CH-Ar), 124.77 (CH-Ar), 122.48 (CH-Ar), 120.58 (CH-Ar), 117.21 (CH-Ar), 112.96 (CH-Ar), 105.59 (CH-Ar) ppm.▪IR (thin film): ν = 3402 (NH-indole), 3321 (NHCO), 3117, 3067, 1667 (br CO), 1651 (CONH), 1589, 1535, 1416, 1331, 1292, 1238, 954, 840, 755, 710 cm^−1^.▪HRMS (ESI, negative mode): *m*/*z* (M^+^-H^+^) 365.09317 (C_23_H_13_N_2_O_3_) requires 365.09262.▪Elemental analysis (C_23_H_14_N_2_O_3_):○Theoretical values (calculated): C, 75.40%; H, 3.85%; and N, 7.65%.○Experimental values (found): C, 75.72%; H, 3.74%; and N, 7.87%.

Compound (**16**), *N*-(9,10-Dihydro-9,10-dioxoanthracen-1-yl)-2-furamide

A mixture of 1-aminoanthraquinone (**11**, 0.40 g, 1.8 mmol) and furan-2-carbonyl chloride (**15**, 0.45 g, 3.5 mmol) was refluxed at 120 °C for 18 h under an air-cooled condenser. Afterward, 15 mL of 1,4-dioxane was added, and the reaction mixture was stirred at room temperature for an additional 24 h. The resulting solid product was purified through filtration, washed with water, and dried. Further purification via recrystallization from chloroform yielded the *title compound* (**16**, 0.18 g, 32%) as a dark red solid. The compound underwent comprehensive characterization, with the following analytical data obtained:▪R*_f_*: 0.67 (CHCl_3_/CH_3_OH, 9:1).▪Melting point: 280 °C (decomposed).▪^1^H-NMR (300 MHz, DMSO-*d_6_*) rotamers: δ = 13.18 (br s, 1H, NHCO), 9.15 (br s, 1H, Ar-H, H3-furan), 8.35–8.10 (m, 2H, Ar-H), 7.94–7.72 (m, 4H, Ar-H), 7.46–7.12 (m, 2H, Ar-H), 6.95–6.61 (m, 1H, Ar-H) ppm.▪^13^C-NMR (DMSO-*d_6_*) rotamers: δ = 192.82 (CO-ketone), 186.44 (CO-ketone), 161.22 (CONH), 148.09, 147.51, 147.22, 136.33, 135.25, 127.62, 126.96, 125.71, 122.59, 118.83, 118.18, 116.91, 113.52, 112.77, 112.56 ppm.▪Elemental analysis (C_19_H_11_NO_4_):○Theoretical values (calculated): C, 71.92%; H, 3.49%; and N, 4.41%.○Experimental values (found): C, 71.55%; H, 3.63%; and N, 4.66%.

### 3.4. Cell Proliferation Study

#### 3.4.1. Cells and Cell Culture Conditions

The human colorectal carcinoma (HCT-116), human breast adenocarcinoma (MCF-7), human chronic myelogenous leukemia (K-562), and human dermal fibroblast normal cell lines were purchased from were purchased from the American Type Culture Collection (ATCC) (Manassas, VA, USA) and stored in liquid nitrogen. Cells were passaged twice weekly upon reaching 70–80% confluency, and only low passage numbers (≤ 10) were used in all experiments [6]. Mycoplasma contamination was monitored using the Lonza MycoAlert™ mycoplasma detection kit. Cells were sub-cultured and maintained with appropriate media: RPMI-1640 medium (Buffalo, New York, USA) was used to grow MCF-7, K-562 and HCT-116 cancer cells, while fibroblast cells were cultured in Iscove’s Modified Dulbecco’s medium (IMDM) (EuroClone S.p.A., Pero, MI, Italy). Both media were supplemented with 10% (*v*/*v*) heat-inactivated fetal bovine serum (FBS) (Gibco, CA, USA), 2 mM of L-Glutamine, 100 units/mL of Penicillin, and 100 μg/mL of Streptomycin (EuroClone, Italy). Cells were maintained at 37 °C in a humidified incubator with 5% CO_2_ (NuAire, Plymouth, MN, USA) [69,70,71].

#### 3.4.2. MTT Assay

The MTT colorimetric assay utilized in this study was previously described [69,70,71,72,73,74] and adapted from Mosmann [34]. The MTT regent [3-(4,5-dimethylthiazol-2-yl)-2,5-diphenyl tetrazolium bromide] was employed to evaluate the antiproliferative activities of carboxamides 4-16 against MCF-7, K-562, HCT-116, and dermal normal fibroblast cell lines. Cells (6–8 × 10^3^ per well, 180 µL) were seeded into 96-well plates (Corning, New York, USA) and allowed to adhere overnight at 37 °C in a humidified 5% CO₂ incubator. Fresh top stock solutions of the test compounds (10 mM in DMSO) were then freshly prepared, followed by serial dilutions in the appropriate culture media before being added to the seeded cells. Control wells received media alone (20 µL per well), ensuring that the final DMSO concentration in the wells did not exceed 1%. Initial screening for potential antiproliferative activity was carried out using two concentration points (10 µM and 50 µM) in triplicate over a 72-hour incubation period. Compounds demonstrating more than 50% inhibition of cell proliferation were further evaluated across a broader concentration range for all tested cell lines. Following 72 h of exposure, MTT solution (2 mg/mL in PBS, 50 µL per well) was added, and plates were incubated for an additional 4 h. The supernatant in each well was then removed, and the resulting formazan crystals were solubilized by adding DMSO (150 µL per well). Absorbance, measured as the optical density (OD), was recorded at 570 nm using a microplate reader (Thermo Fisher Scientific, Waltham, MA, USA). The detected signal intensity corresponds to metabolic activity, which directly correlates with the viable cell number.

#### 3.4.3. Statistical Analysis

All experiments were performed in triplicate, and the results are presented as the mean ± standard deviation (SD). The IC_50_ values for antiproliferative activity were calculated using nonlinear regression analysis in GraphPad Prism software (GraphPad Prism version 9.0.0 for Windows, GraphPad Software, San Diego, California USA). Additionally, a two-way ANOVA test was conducted to analyze significant variations in IC_50_ values, with a significant level set at α = 0.05.

### 3.5. Molecular Modeling and Cheminformatics Methods

#### 3.5.1. Protein Preparation

The X-ray crystal structures of human topoisomerase–DNA (PDB ID: 5ZRF) [36], phosphoinositide 3-kinase (PI3Kα) (PDB ID: 4L23) (resolution = 2.50 Å) [37], and epidermal growth factor receptor (EGFR) (PDB ID: 3W32) (resolution = 1.80 Å) complex [38] were downloaded from the Protein Data Bank (PDB). Protein preparation was carried out using MAESTRO software (version 2022.2) [39], which involved filling in missing residues, capping the N- and C-termini, minimize hydrogen atoms, and optimizing the hydrogen bond network. Furthermore, side-chain perturbations were applied to reduce steric clashes and enhance structural accuracy.

#### 3.5.2. Ligand Structure Preparation

The validated ligand molecules were constructed using the Build panel in MAESTRO, based on the template of the co-crystallized ligand (EVP) in the 5ZRF structure. Further energy minimization and conformational refinement of the ligands were performed using the LigPrep module in MAESTRO [39] to ensure optimal geometry and stability for subsequent docking studies.

#### 3.5.3. Induced-Fit Docking (IFD)

The co-crystallized ligands EVP/5ZRF, X6K/4L23, and W32/3W32 were designated as centroids within their respective binding sites. To ensure optimal ligand–receptor interactions, the van der Waals scaling factors for both the receptor and ligand were adjusted to 0.5, allowing for appropriate relaxation and flexibility during docking. All other docking parameters were maintained at their default settings. The ligand pose with the highest XP Glide binding score was selected for further analysis. Docking scores were reported in kcal/mol, with more negative values indicating stronger binding affinities.

#### 3.5.4. Molecular Descriptors

All molecular structures were sketched in ChemDraw Professional 16.0 [75] and saved in SDF file format. The structures were then standardized according to the methods described by Hajjo et al. [76]. Subsequently, 2D molecular descriptors were calculated using alvaDesc software (version 1.0.22) from Kode Cheminformatics to facilitate further computational analysis [66].

#### 3.5.5. Principal Component Analysis (PCA)

Principal component analysis (PCA) was conducted on all synthesized active molecular structures utilizing the calculated 2D alvaDesc molecular descriptors. This analysis was performed using alvaDesc software (version 1.0.22) from Kode Cheminformatics, enabling a systematic exploration of molecular diversity, the identification of structural patterns, and the assessment of correlations among key molecular properties [66].

## 4. Conclusions

This study successfully designed, synthesized, and evaluated a series of *N*-substituted 1*H*-indole-2-carboxamides, demonstrating their potential as novel anticancer agents. The synthesized compounds exhibited significant antiproliferative activity across various cancer cell lines, with a notable selectivity for cancerous cells over normal fibroblasts. Among the compounds, 4, 12, and 14 were particularly effective against the K-562 cell line, showing strong cytotoxicity, while compound 10 exhibited the highest potency against the HCT-116 cell line. The MCF-7 cells showed moderate sensitivity, highlighting the compounds’ broad applicability across different types of cancer. Molecular modeling and computational studies further enhanced our understanding of the mechanisms of action, revealing the critical binding interactions between these derivatives and key oncogenic targets, including topoisomerase–DNA, PI3Kα, and EGFR. These computational predictions were consistent with experimental findings, providing a strong basis for the observed anticancer activity. The data suggest that these derivatives effectively disrupt the function of these targets, which are integral to cancer cell proliferation and survival, thereby presenting a multi-target approach to cancer treatment. Furthermore, the principal component analysis of the molecular descriptors revealed that the synthesized compounds occupy a diverse space of physicochemical properties, indicating their potential for drug-like and lead-like characteristics. Among them, compound 10 emerged as the most promising candidate, displaying both excellent drug-like and lead-like properties, positioning it as a favorable candidate for further development.

This work not only supports the potential of these indole carboxamide derivatives as anticancer agents but also underscores the importance of combining experimental and computational strategies in drug discovery. The strong correlation between in vitro results and computational predictions demonstrates that these compounds may offer a novel therapeutic approach with fewer off-target effects compared to traditional chemotherapies. The findings from this study provide a foundation for future research aimed at optimizing the efficacy, specificity, and pharmacokinetic properties of these derivatives, with the ultimate goal of advancing their clinical development as effective and less toxic anticancer agents. This work contributes significantly to the ongoing global efforts to discover and develop new, effective, and less toxic anticancer therapies. By focusing on selective inhibition of key molecular pathways involved in cancer progression, these compounds may pave the way for more personalized and targeted cancer treatments in the future.

### Limitation and Future Investigations

Although the synthesized *N*-substituted indole-2-carboxamides demonstrated promising anticancer activity in vitro, we acknowledge the necessity of in vivo validation to further substantiate their therapeutic potential. Preliminary in vivo studies, including xenograft models to assess pharmacokinetics and toxicity profiles, are currently underway and will be reported in a subsequent publication. These future investigations will provide deeper insights into the compounds’ efficacy, safety, and translational relevance. We believe this continued work will strengthen the preclinical development of these derivatives as viable anticancer candidates.

## Data Availability

The datasets used and analyzed during the current study are available upon reasonable request from the corresponding author. Please note that due to the ongoing experiment’s requirements in the later stages, the data will be released at this time.

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
