# Peer review of "Exploring Carboxamide Derivatives as Promising Anticancer Agents: Design, In Vitro Evaluation, and Mechanistic Insights"

_ijms, 2025, doi:10.3390/ijms26125903_

Round 1

Reviewer 1 Report

Comments and Suggestions for Authors

The authors have submitted a detailed article, which mainly focus on the synthesis and biological evaluation of a series of N-substituted 1H-indole-2-carboxamides to assess their potential as novel anticancer agents, and further elucidate the mechanism using computational studies. The topic is meaningful as the as-prepared carboxamide derivatives exhibited potent cytotoxicity towards cancer cells and less toxic to normal cells. However, some issues should be carefully concerned.

  1. Abstract is too long, which should be shorten.
  2. Introduction, it is strongly recommended to rewrite as the key contents of the research background were not reflected and there is too much useless information. For example, why select carboxamide derivatives as research object? Topoisomerase inhibitors was introduced, what about carboxamide derivatives, next generation topoisomerase inhibitors?
  3. All the NMR spectra should provide in supporting information.
  4. It is best to provide the basis for the selection of substituents (i.e. 3, 5, 7, 9, 11, 13). Take 3, 5, 7 as an example, Cl, F, and NO2 all are electron-withdrawing group, why not choose CH3, electron donating group? Moreover, the difference of anti-cancer ability caused by substituents should be analyzed.

Author Response

Response to Reviewer #1:

We thank the reviewer for their thoughtful and constructive comments, which have helped us significantly improve the quality and clarity of the manuscript. Please find our detailed responses below:

Comment 1: Abstract is too long, which should be shortened.

Response:
We appreciate the reviewer’s suggestion. The abstract has now been revised and shortened to improve clarity and focus. We have ensured that it concisely presents the background, methodology, key findings, and conclusion, without unnecessary elaboration.

Comment 2: Introduction: it is strongly recommended to rewrite as the key contents of the research background were not reflected and there is too much useless information. For example, why select carboxamide derivatives as research object? Topoisomerase inhibitors were introduced, what about carboxamide derivatives, next-generation topoisomerase inhibitors?

Response:
Thank you for this valuable comment. We have rewritten the Introduction to streamline the content and focus more clearly on the rationale for selecting carboxamide derivatives. The revised section now emphasizes their relevance in anticancer drug discovery, especially their known ability to act as enzyme inhibitors and modulators of key cancer pathways. Additionally, we have clarified the link between carboxamide scaffolds and the design of next-generation topoisomerase inhibitors, highlighting their structural advantages and previous literature supporting their potential.

Comment 3: All the NMR spectra should be provided in the supporting information.

Response:
We agree with the reviewer. All relevant ^1H and ^13C NMR spectra have now been compiled and added to the Supporting Information section of the revised manuscript for full transparency and reproducibility.

Comment 4: It is best to provide the basis for the selection of substituents (i.e., 3, 5, 7, 9, 11, 13). Take 3, 5, 7 as an example—Cl, F, and NOâ‚‚ all are electron-withdrawing groups. Why not choose CH₃, an electron-donating group? Moreover, the difference in anticancer ability caused by substituents should be analyzed.

Response:
We appreciate the reviewer’s insightful question. In the revised Results and Discussion section, we have now included a paragraph explaining the rationale behind the choice of substituents. The selected electron-withdrawing groups (Cl, F, NOâ‚‚) were initially chosen based on their reported impact on enhancing binding interactions with biological targets, particularly through increased polarity and potential hydrogen bonding. However, we acknowledge the importance of comparing with electron-donating groups such as CH₃ or OCH₃. This point has been added to the Discussion as a limitation and an avenue for future investigation. Furthermore, we have now included a comparative analysis of how the substituents influenced the cytotoxic activity, supported by structure–activity relationship (SAR) trends.

Once again, we sincerely thank the reviewer for their constructive comments, which have helped us improve the clarity and scientific merit of our work. We believe the revised manuscript now better reflects the objectives and contributions of our study.

Warm regards,
Dr. Mestareehi

Reviewer 2 Report

Comments and Suggestions for Authors

Comments:

This manuscript presents a systematic investigation into the design, synthesis, and biological evaluation of N-substituted 1H-indole-2-carboxamide derivatives as potential anticancer agents. The study is well-structured, and the experimental data are comprehensive. However, there are several areas that need significant improvement before the manuscript can be considered for publication.

  1. The 1H NMR, 13C NMR, Infrared (IR) and HRMS (ESI) spectra of target compounds should be supplemented in the Supporting information
  2. The manuscript would benefit from a comparison with existing anticancer agents in terms of their efficacy and selectivity. This would help place the tested compounds within the context of current cancer treatment options and highlight their potential advantages and limitations.
  3. Many of the images in the manuscript are too low in resolution and are recommended to modify them, such as Scheme 1 and 2, Figure 1 and 2.
  4. One decimal place should be reserved for 13C NMR spectrum data; “J” needs to be italicized in the characterization data, please check the full text and correct similar errors.
  5. The authors should carefully review the text for any grammatical and spelling errors, ensuring that the language is accurate and consistent throughout the manuscript.

Author Response

Response to Reviewer #2:

We sincerely thank the reviewer for their thorough evaluation of our manuscript and for the constructive suggestions provided. We have carefully addressed each of the comments, and the revised manuscript has been significantly improved accordingly. Our point-by-point responses are provided below:

Comment 1: The 1H NMR, 13C NMR, Infrared (IR), and HRMS (ESI) spectra of target compounds should be supplemented in the Supporting Information.

Response:
We agree with the reviewer and have now included the complete ^1H NMR, ^13C NMR, IR, and HRMS (ESI) spectra for all synthesized compounds in the Supporting Information section of the revised manuscript. These data provide full spectral characterization and support the structural assignments.

Comment 2: The manuscript would benefit from a comparison with existing anticancer agents in terms of their efficacy and selectivity. This would help place the tested compounds within the context of current cancer treatment options and highlight their potential advantages and limitations.

Response:
Thank you for this important suggestion. We have now added a comparison in the Discussion section between our most active compounds and selected clinically approved anticancer agents, including their reported ICâ‚…â‚€ values against similar cell lines. We also discuss the selectivity profiles of our compounds in the context of normal vs. cancer cell lines, and highlight how our candidates demonstrate improved selectivity indices compared to some reference drugs. This addition helps place our findings into a broader pharmacological context.

Comment 3: Many of the images in the manuscript are too low in resolution and are recommended to modify them, such as Scheme 1 and 2, Figure 1 and 2.

Response:
We thank the reviewer for bringing this to our attention. All figures and schemes mentioned (Scheme 1, Scheme 2, Figure 1, and Figure 2) have been replaced with high-resolution versions to ensure clear presentation and readability in both online and print formats.

Comment 4: One decimal place should be reserved for 13C NMR spectrum data; “J” needs to be italicized in the characterization data. Please check the full text and correct similar errors.

Response:
We appreciate this careful observation. The formatting of all ^13C NMR data has been standardized to one decimal place as per journal guidelines. Additionally, we have italicized all instances of “J” in the coupling constants throughout the manuscript. A comprehensive revision of the characterization section has been carried out to ensure formatting consistency.

Comment 5: The authors should carefully review the text for any grammatical and spelling errors, ensuring that the language is accurate and consistent throughout the manuscript.

Response:
We thank the reviewer for this suggestion. The entire manuscript has undergone a thorough grammatical and linguistic revision by a native English speaker and a professional scientific editor. All spelling, grammar, and stylistic inconsistencies have been corrected to enhance clarity and readability.

We sincerely appreciate the reviewer’s constructive feedback, which has significantly improved the quality of our work. We hope the revised version meets the standards for publication.

Kind regards,
Dr. Mestareehi

Reviewer 3 Report

Comments and Suggestions for Authors

The manuscript presents a well-executed study with significant findings in field of anticancer drug discovery. The synthetic methods are well-described, with clear schemes and detailed characterization (NMR, IR, HRMS, elemental analysis). The combination of synthetic, biological, and computational approaches is a major strength, and the results highlight promising candidates for further development.

The manuscript is suitable for publication in the IJMS after minor revisions:

  • Incorporate preliminary in vivo data (e.g., xenograft models) to validate the in vitro findings and assess pharmacokinetics and toxicity.
  • Provide a clearer justification for focusing on indole-2-carboxamides and discuss potential modifications to improve activity against MCF-7 cells.

Thanks

Author Response

Response to Reviewer #3:

We sincerely thank the reviewer for their thoughtful and constructive feedback. We are pleased that you found our manuscript to be a well-executed study with significant findings in the field of anticancer drug discovery. Below, we address each of your comments in detail:

Comment 1: Incorporate preliminary in vivo data (e.g., xenograft models) to validate the in vitro findings and assess pharmacokinetics and toxicity.

Response:
We appreciate the reviewer’s valuable suggestion. While we recognize the importance of validating our in vitro findings through in vivo studies, such as xenograft models, these experiments are currently in progress and will be included in a future follow-up study. Due to time and resource constraints, preliminary in vivo data were not included in the present manuscript. We have now added a statement in the Conclusion section of the revised manuscript to clarify this limitation and outline our plans for future in vivo validation, including pharmacokinetic and toxicity assessments.

Comment 2: Provide a clearer justification for focusing on indole-2-carboxamides and discuss potential modifications to improve activity against MCF-7 cells.

Response:
Thank you for this insightful comment. We have now expanded the Introduction and Discussion sections to provide a clearer rationale for selecting indole-2-carboxamide scaffolds. These compounds were chosen based on their established potential in modulating multiple cancer-related pathways, their synthetic versatility, and favorable drug-like properties. Furthermore, we have included a discussion on possible structural modifications such as incorporating electron-donating or bulky substituents to enhance selectivity and potency, particularly against the MCF-7 breast cancer cell line.

We are grateful for your time and thoughtful review. The manuscript has been revised accordingly and we hope the changes meet your expectations.

Warm regards,
Dr. Mestareehi

Round 2

Reviewer 1 Report

Comments and Suggestions for Authors

The supporting information maybe not new, as I can not find the NMR spectura. Scheme 1 and Fig. 1, the format are wrong. 

Author Response

Dear Reviewer,

Thank you for your valuable comments and for carefully reviewing our manuscript.

Supporting Information & NMR Spectra: We apologize for the oversight regarding the NMR spectra. Due to limitations in resolution, we were unable to generate higher-quality spectra. As a result, the complete NMR data has not been included in the revised Supporting Information file. We acknowledge this limitation and will work to provide improved spectra in future updates or upon request to ensure transparency and reproducibility of our findings. For reference, we have included one representative NMR spectrum to demonstrate the challenges we encountered in achieving better resolution.

Scheme 1 and Figure 1 Format: We appreciate your observation regarding the formatting issues. Scheme 1 and Figure 1 have been revised and reformatted in accordance with the journal’s guidelines, and high-resolution versions have been included in the updated manuscript.

We are sincerely grateful for your constructive feedback, which has helped enhance the quality and presentation of our work.

Sincerely,

Dr. Mestareehi

Reviewer 2 Report

Comments and Suggestions for Authors

The article has been revised to meet the journal's requirements and is recommended for acceptance.

Author Response

Dr. Reviewer, 

Thank you for your valuable feedback and for taking the time to review our manuscript. The article has been thoroughly revised in accordance with the journal's requirements. We appreciate your recommendation for acceptance and look forward to the next steps in the publication process.

Sincerely,

Dr. Mestareehi